# Mathematical models of drug-resistant tuberculosis lack bacterial heterogeneity: A systematic review

Naomi M. Fuller[1,2,3,4]*, Christopher F. McQuaid[1,2,3,4], Martin J. Harker[1,2,3,4], Chathika K. Weerasuriya[1,2,3,4], Timothy D. McHugh[5], Gwenan M. Knight[1,2,3,4]

1 Department of Infectious Disease Epidemiology, Faculty of Epidemiology and Population Health, London School of Hygiene and Tropical Medicine, London, United Kingdom, 2 Centre for Mathematical Modelling of Infectious Diseases, London School of Hygiene and Tropical Medicine, London, United Kingdom, 3 Antimicrobial Resistance Centre, London School of Hygiene and Tropical Medicine, London, United Kingdom, 4 Tuberculosis Centre, London School of Hygiene and Tropical Medicine, London, United Kingdom, 5 UCL Centre for Clinical Microbiology, Division of Infection & Immunity, Royal Free Campus, University College London, London, United Kingdom

* naomi.fuller@lshtm.ac.uk

**Data Availability Statement:** All relevant data are within the manuscript and its Supporting Information files.

## Abstract

Drug-resistant tuberculosis (DR-TB) threatens progress in the control of TB. Mathematical models are increasingly being used to guide public health decisions on managing both antimicrobial resistance (AMR) and TB. It is important to consider bacterial heterogeneity in models as it can have consequences for predictions of resistance prevalence, which may affect decision-making. We conducted a systematic review of published mathematical models to determine the modelling landscape and to explore methods for including bacterial heterogeneity. Our first objective was to identify and analyse the general characteristics of mathematical models of DR-mycobacteria, including *M. tuberculosis*. The second objective was to analyse methods of including bacterial heterogeneity in these models. We had different definitions of heterogeneity depending on the model level. For between-host models of mycobacterium, heterogeneity was defined as any model where bacteria of the same resistance level were further differentiated. For bacterial population models, heterogeneity was defined as having multiple distinct resistant populations. The search was conducted following PRISMA guidelines in five databases, with studies included if they were mechanistic or simulation models of DR-mycobacteria. We identified 195 studies modelling DR-mycobacteria, with most being dynamic transmission models of non-treatment intervention impact in *M. tuberculosis* (n = 58). Studies were set in a limited number of specific countries, and 44% of models (n = 85) included only a single level of "multidrug-resistance (MDR)". Only 23 models (8 between-host) included any bacterial heterogeneity. Most of these also captured multiple antibiotic-resistant classes (n = 17), but six models included heterogeneity in bacterial populations resistant to a single antibiotic. Heterogeneity was usually represented by different fitness values for bacteria resistant to the same antibiotic (61%, n = 14). A large and growing body of mathematical models of DR-mycobacterium is being used to explore intervention impact to support policy as well as theoretical explorations of resistance dynamics.

**Funding:** This research and NMF was funded by the Biotechnology and Biological Sciences Research Council through the London Interdisciplinary Doctoral Training Programme (BBSRC LIDO, https://www.lido-dtp.ac.uk) at the London School of Hygiene and Tropical Medicine (LSHTM) in partnership with University College London (UCL), Grant code - BB/M009513/1. CFM was funded for other work by Bill and Melinda Gates Foundation (TB MAC OPP1135288, INV-059518, https://www.gatesfoundation.org) and Unitaid (20193–3-ASCENT, https://unitaid.org/calls/#en). CKW was supported by a grant from the Bill and Melinda Gates Foundation (INV-001754, https://www.gatesfoundation.org). GMK was supported by Medical Research Council UK, https://www.ukri.org/opportunity/career-development-award/ (MR/ W026643/1). The views expressed are those of the authors and not necessarily those of the BBSRC, LIDO, LSHTM or UCL. The funders had no role in study design, data collection and analysis, decision to publish, or preparation of the manuscript.

**Competing interests:** The authors have declared that no competing interests exist.

However, the majority lack bacterial heterogeneity, suggesting that important evolutionary effects may be missed.

## Author summary

The emergence of drug-resistant tuberculosis (DR-TB), where the causative bacterium *Mycobacterium tuberculosis* is resistant to key antibiotics such as rifampicin and isoniazid, poses a significant threat to TB control efforts. To gain a broader understanding of the challenges surrounding DR-TB, mathematical models are increasingly being employed to estimate the impact of interventions, effectiveness of treatment, and to predict the evolution of drug-resistance. However, pragmaticism surrounding model construction often means that important aspects, such as bacterial heterogeneity, are overlooked. We undertook a systematic review of the existing DR-mycobacterium modelling literature, with the specific aim of capturing methods for including bacterial heterogeneity. Our analysis revealed that most models of drug-resistance in mycobacteria primarily focus on intervention strategies and cost-effectiveness analyses, with minimal attention to bacterial heterogeneity. Where heterogeneity is included it mostly consisted of different fitness costs for resistance.

## Introduction

Drug-resistant (DR-) strains of *Mycobacterium tuberculosis* (*M. tuberculosis*) are an urgent threat to the control of tuberculosis disease (TB) globally. For TB, the backbone antibiotics of standard therapy are rifampicin and isoniazid. In 2021, multidrug-resistant (combined rifampicin and isoniazid resistance) or rifampicin-resistant tuberculosis (MDR/RR-TB) caused an estimated 450,000 cases globally [1].

Routinely collected antimicrobial resistance (AMR) data use microbiological definitions of resistance, which are guided by threshold cut-offs for phenotypic resistance, resulting in discrete categorisations. For TB, these categorisations are further grouped with strains being classified as drug-susceptible (DS-), multidrug- or rifampicin-resistant- (MDR/RR-), pre-extensively-drug (pre-XDR) resistant (MDR plus resistance to a fluoroquinolone) or XDR-resistant (MDR plus resistance to a fluoroquinolone and a Group A drug) [1]. The MDR/RR grouping is based on the knowledge that isoniazid resistance is commonly acquired prior to rifampicin resistance and the wider prevalence of rifampicin-resistance testing through genotypic testing, making clinical management of RR- and MDR-TB similar [2,3]. These definitions are sufficient for patient care decision-making that does not need to account for the spectrum of phenotypic resistance levels (for example, those below the threshold for successful treatment) or any other bacterial characteristics (such as types of resistance-conferring mutations). However, bacterial populations are often highly diverse with a spectrum of characteristics. Hence, resistance categories will also have a high degree of bacterial heterogeneity, such as variation in transmission fitness between strains with the same phenotypic resistance, which affects the rate at which *M. tuberculosis* spreads between individuals.

Several important insights into the evolution of DR-TB, its emergence and spread, and the control of resistant bacteria more broadly have been generated by mathematical models. Some examples are the predominance of primary rather than acquired resistance, the effectiveness of TB surveillance for controlling DR-TB, and the potential impact of controlling HIV on

reducing TB transmission [4–7]. Most mathematical models of AMR have typically adopted binary (*e.g.* resistant versus susceptible) categorisations. When bacterial heterogeneity is included in mathematical models, the predicted public health outcomes can be different from those when bacterial heterogeneity is ignored [8]. We may lose subtlety in model outputs when modelling antibiotic treatment as a selective pressure if the traits allowing for bacterial heterogeneity are not included. Models may miss key dynamics, such as competition between strains and antibiotic effectiveness against strains with varying resistance levels, and be at risk of incorrectly predicting the effectiveness of a treatment intervention. As Trauer et al. (2018) point out, strain diversity, virulence and fitness costs have implications for the trajectory of drug resistance in TB [9]. Decisions as to what to include in a model will depend on the questions being asked, the selective pressures modelled, and the time-frame studied. Assessing this balance in model design between detailed and generalised parameters to allow a pragmatic approach for public health interventions can often prove challenging. Hence, assessing the extent to which bacterial heterogeneity has been included in existing models that predict intervention impact for DR-TB control is highly important.

Previous systematic reviews have explored the landscape of mathematical models of AMR [7,10] and TB [11–14], with up to 43 DR-TB transmission and 52 within-host studies being found prior to 2016. To our knowledge, only one expert review from 2009 focused on mathematical models of DR-TB [4], emphasising the useful insights from modelling but also highlighting important knowledge gaps in the economics, biological impact of mutations and ability to control DR-TB. To date, there is little evidence on how bacterial heterogeneity is incorporated into DR-TB models and little evidence of the effect this would have on model outcomes.

Mycobacteria predominantly develop antibiotic resistance via mutation [15], resulting in different patterns of resistance dynamics to other bacterial genera. Mycobacterial species other than *M. tuberculosis* can often be used as experimental or theoretical models for *M. tuberculosis* and are also responsible for a clinical burden [16–18]. They are often used to understand the resistance dynamics of *M. tuberculosis* [19,20].

We aimed to support future modelling of interventions against DR-TB by systematically surveying the characteristics of mathematical models of mycobacteria, of which we expect the *M. tuberculosis* species to dominate due to its substantial clinical burden. Our secondary objective was categorising the amount and type of bacterial heterogeneity included in mathematical models of DR-mycobacteria. We envisaged two broad settings of papers to be included in this review, within-host and between-host transmission models. This was noted by Cohen et al. (2009), a previous review of the DR-TB modelling literature [4], where "between-host" models refer to models on the human population scale. Since 2009, there has been an increase in models of bacterial populations set in the laboratory. As the populations captured will be similar to within-host models, we combined laboratory models and within-host models and collectively called them "bacterial population" models.

The aims, dynamics and model structure of between-host models differ considerably from bacterial population models, namely by transmission of the pathogen and populations included, making them difficult to compare. Therefore, we defined heterogeneity differently for bacterial populations and between-host models to compare methods within these categories and gain a clearer picture of bacterial heterogeneity modelling. At the between-host level, we were interested in capturing those models that went beyond capturing resistance phenotypes but included any added dimension of bacterial variation, including what may affect survival, such as fitness effects. Models of bacterial populations that captured any resistance variation were included; distinct populations of resistant bacteria needed to be modelled, which differed in their parameter values (e.g. growth rate or mutation rate).

## Methods

Our review consisted of two stages of selection and data analysis. In Stage 1 of the review, our aim was to identify and analyse the general characteristics of mathematical models pertaining to drug-resistant (DR-) mycobacteria, such as model type and aim. In Stage 2 of the review, our focus was to identify mathematical models of DR-mycobacteria that specifically incorporated the concept of bacterial heterogeneity, as elucidated by the definition in the inclusion and exclusion criteria section.

### Search strategy

The systematic review was designed and conducted following the PRISMA reporting protocol to search and review mathematical modelling papers of DR-mycobacteria [21]. The search terms consisted of those relevant to [1] "mycobacteria", [2] "mathematical modelling", and [3] "antibiotic resistance" (S1 Text). The search was conducted in five databases (Medline, Embase, Global Health, Web of Science and Scopus) initially on January 22nd, 2021, and then repeated on April 1st, 2022. Duplicates were removed before screening.

### Inclusion and exclusion criteria

The screening process of the papers adhered to predefined inclusion and exclusion criteria (Table 1). Initially, the titles and abstracts of the papers were screened to identify mathematical models specifically pertaining to DR-mycobacteria, followed by a full-text screening for inclusion in Stage 1. Finally, another round of full-text screening was carried out on the remaining papers to identify those appropriate for Stage 2 of the study.

Mathematical models were defined as mechanistic models or simulation models reproducing a mathematically described scenario of DR-mycobacteria or of individuals carrying DR-mycobacteria. We excluded statistical analyses, such as regression models or risk analysis; molecular modelling (those focused on molecular structure of chemical compounds) or those only focused on drug development; models of drug-resistance that only used mycobacteria as an example or discussion point unless results for DR-mycobacteria were specifically included.

We split models into two groupings: "between-host" and "bacterial population" models, with the differences in their model scale, structure, and aims, resulting in a different bacterial heterogeneity definition. A "between-host" model was classed as a heterogenous model when strains infecting a human population resistant to the same drug varied in another characteristic such as fitness, rates of compensatory mutation evolution or associated treatment recovery rates. These characteristics were extracted during the full-text extraction stage. "Bacterial

**Table 1. Inclusion and exclusion criteria used for title and abstract, stage 1 and stage 2 screening.**

| Inclusion Criteria | | Exclusion Criteria |
|---|---|---|
| **Stage 1** | **Stage 2** | |
| Mathematical model capturing at least one mycobacterial species | Mathematical model of resistance in mycobacteria with a heterogeneous bacterial component | Reviews, opinion pieces, editorials, letters, model comparison exercises, conference abstracts |
| Mathematical model with a population of antibiotic-resistant bacteria or individuals carrying resistant bacteria | | Molecular modelling, drug development, genetic pathways, genetic evolution models, statistical analysis only |
| English language | | Pharmacokinetics/Pharmacodynamics (PK/PD) model with no resistant bacterial population |
| | | Models that use data but do not produce DR-mycobacteria results |
| | | Animal (non-human) host |

population" models included both within-host and models of bacterial populations capturing dynamics measured in laboratory or experimental conditions. A bacterial population model was classed as a heterogeneous model when there were distinct resistant strains captured which had different parameter values such as fitness, mutation rates and metabolic states. These parameter differences were extracted during the full-text extraction stage.

## Selection and extraction: Stage 1

Title and abstract screening were performed for every paper by at least two authors (NMF, GMK, CFM, MJH and CKW) to determine if the paper likely included a mathematical model of DR-mycobacteria. High-level data extraction from these screened papers that continued to match the criteria for Stage 1 upon full-text screening provided a landscape analysis of DR-mycobacteria models. DR-mycobacteria models can address multiple aims with various methods, but they will have a common theme, such as parameter estimation or evaluation of the impact of interventions. We extracted information from the models to categorise and classify them into five categories, focusing on the main theme of the model. 1) model setting (such as geographic location), 2) model aims (7 categories of; non-treatment *in*terventions that did not explore antibiotic usage (with and without cost-effectiveness), treatment interventions (with and without cost-effectiveness), parameter estimation, burden estimation or theoretical), 3) model type (7 categories of; bacterial dynamics, decision analytic, PK/PD, state transition (with and without a statistical component) or transmission (with or without an operational or state transition component), 4) mycobacterial species and 5) resistance classifications (such as MDR or XDR) (S2 Text). We extracted resistance classifications based on what the authors defined in their papers, as current resistance definitions are continuously updated. A resistance class is defined as a model stratification whereby strains (or the populations including them) are grouped across multiple antibiotic resistances (i.e. MDR could here be a single "resistance class" but represents resistance to multiple antibiotic agents). We only extracted which antibiotics were modelled in papers if their resistance was also considered. This extraction was performed by NMF and GMK, with discussions to resolve any conflicts.

## Selection and extraction: Stage 2

For Stage 2, full-text screening of the Stage 1 papers was performed by three authors (NMF, GMK, CFM) to determine the models with bacterial heterogeneity, with subsequent discussions and consensus to resolve any discrepancies. NMF performed full-text extraction and data analysis of the extracted data from these papers (S2 Table). Stage 2 extracted data on the methods used to model heterogeneity, types of heterogeneity included, data sources and the effect of resistance inclusion (such as resistance effects on disease progression) (S3 Text).

## Results

After the removal of duplicates, 3,180 papers were identified (Fig 1). Following a title and abstract screening, 372 papers remained for full-text screening. 195 papers were found to fulfil our Stage 1 criteria having a model of DR-mycobacteria strains (S1 Table). Of these papers, only 23 were found to meet the requirements of bacterial heterogeneity in mathematical models of DR-mycobacteria (S2 Table).

## Stage 1 Results: DR-mycobacteria model landscape

Most models of mycobacteria were of *M. tuberculosis* (190 papers/97%) with HIV (59 papers) and diabetes mellitus (5 papers) often included. There was a rapid increase in the number of papers published on DR-mycobacterium from 2005 onwards (S1 Fig).

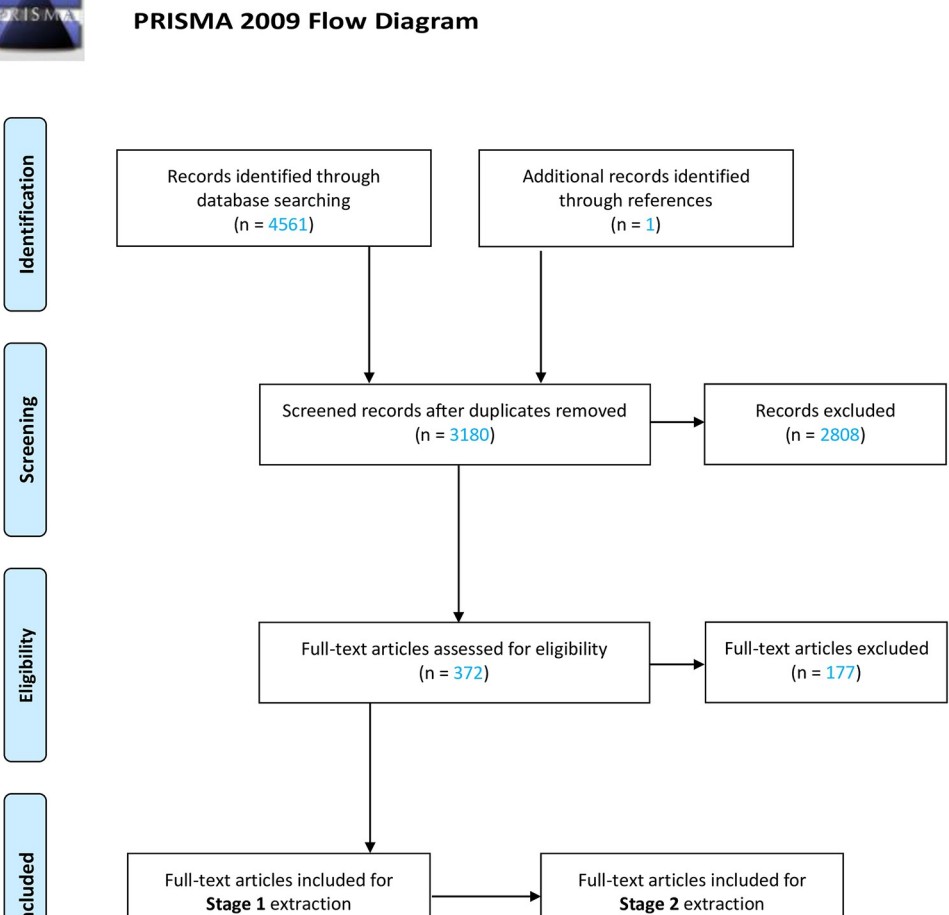

**Fig 1. PRISMA flow diagram outlining the systematic selection of studies to include in the analysis.**

## Settings captured

119 papers aimed to model a specific geographical location, typically at the national level (Fig 2A and S3 Table). This reflects the settings with the highest MDR-TB incidence but also highlights some countries that are not being focused on (Fig 2B). Of the 117 papers, 82 covered a single national analysis and 35 covered different countries. Other geographical locations included 7 models with a global focus, whilst 6 models covered regions with 4 models of Southeast Asia [22–24], and 1 of Eastern Europe [25] and 1 of the Asia-Pacific [26].

## Model aims and types

Of the seven distinct categories of study aim found (Fig 3), non-treatment interventions without cost-effectiveness considered (n = 45, 23%) was the most common. Transmission models (n = 129, 67%) were the most common model type used for all model aims, except for "treatment interventions with cost-effectiveness", which mostly used state transition models (Fig 3). As would be expected, PK/PD models were used almost exclusively for "treatment interventions", with one model being used for parameter estimation. Six models used a combination of

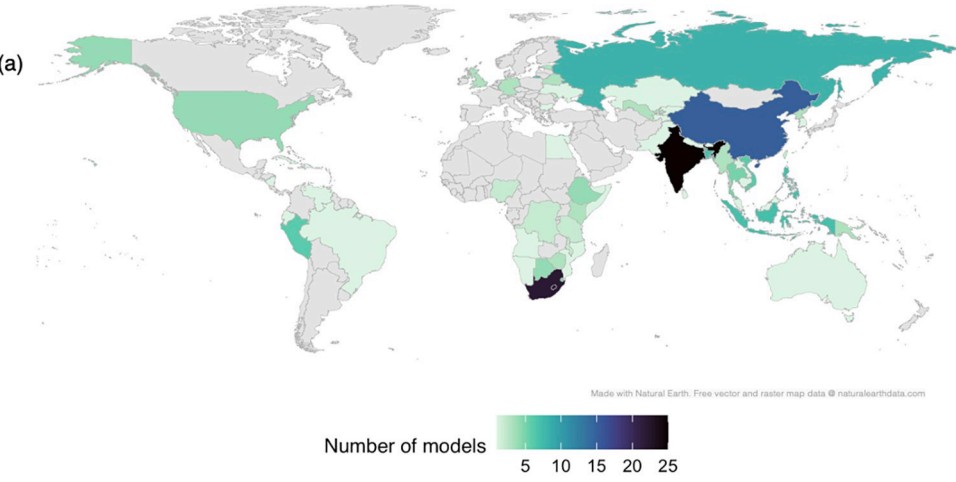

**Fig 2. Uneven geographical distribution of mathematical models of DR-mycobacteria (a) and correspondence with MDR/RR-TB incidence (b).** (a) Countries captured in models of DR-mycobacteria. Note: some models include outputs for multiple countries, therefore this image represents all countries modelled, not the total number of models. (b) From the WHO Global Tuberculosis Report 2022 [1], the 10 countries with the highest estimated MDR/RR-TB incidence are given with number of models in brackets. The colours in the table match the corresponding colours of the country in part (a). Map layer made with Natural Earth, free vector and raster map data @ naturalearthdata.com.

methods: transmission and state transition [27,28], transmission and operational [29] and state transition and statistical [30–32]. "Bacterial dynamics" type models were used for "treatment interventions", "theoretical" and "parameter estimation" aims only. "Decision analytic" type models were used for all aims other than "theoretical" and "parameter estimation".

## Resistance categories

Most models of DR-mycobacteria capture resistance to fewer than three antibiotics. Six models considered all possible combinations of resistance to several antibiotics ('*', Fig 4). Of 16 models to capture four or more resistances at once, 11 of these models included antibiotic resistance as stepwise accumulation of resistance [22,30,33–41] and 5 models only included mono-resistance of resistance to multiple antibiotics [20,42–45].

Overall, for stage 1, most models included a resistance class of MDR/RR-TB (129 papers/ 67%, Fig 4) with 85 models that chose to model only a single resistance class of MDR/RR-TB alongside DS-TB (Fig 4). 40/195 models included isoniazid resistance (Fig 4) with 27/40 also including MDR/RR-TB. 21/195 models included rifampicin resistance separate from MDR with 15/21 including isoniazid and rifampicin resistance as mono-resistances that developed into MDR with 6/15 models including the development of XDR-TB. Of 18 models that modelled XDR, 16 included MDR/RR, while two did not [46,47]. Out of the first-line antibiotics used to treat TB, isoniazid (n = 40) and rifampicin (n = 27) resistance were modelled the most, followed by pyrazinamide (n = 8) and then ethambutol (n = 5) resistance. Pyrazinamide

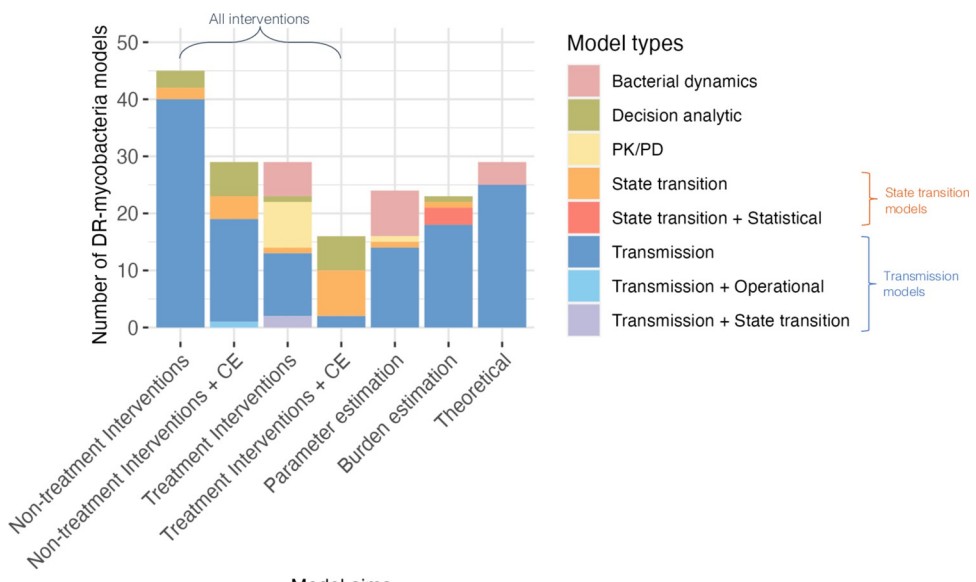

**Fig 3. Model aims broken down by model types (colours) highlights transmission models are the most used for DR-mycobacteria modelling.** The model type (colours) definitions can be summarised as follows: [1] Bacterial dynamics: Capture bacterial populations without considering between-host transmission. [2]; Decision analytic: Track cohorts of human individuals through treatment or diagnostic pathways without ongoing transmission. [3] Pharmacokinetic/pharmacodynamic (PK/PD): Focus on drug concentrations and their effects in vivo, incorporating parameters related to bacterial populations. [4] State Transition: Involve individuals or populations transitioning between different disease states, with the force of infection as a static input parameter. [5] Statistical: inference-based models of collected or population data. [6] Transmission: Dynamically account for the spread of bacteria between individuals or populations. [7] Operational models: simulation of patient pathways and treatment or diagnostic procedures. The model aim (x axis) definitions can be summarised as follows: (1) Non-treatment Interventions: Model the impact of interventions not related to changes in antibiotic usage or treatment without considering economic aspects. (2) Non-treatment Interventions + cost-effectiveness: Model the impact of interventions not related to changes in antibiotic usage or treatment while considering their economic impact. (3) Treatment interventions: Model interventions related to changes in antibiotic usage. (4) Treatment interventions + cost-effectiveness: Model interventions related to changes in antibiotic usage while considering their economic impact. (5) Parameter estimation: Estimate parameters by comparing to data, trends, or varying model structures or components. (6) Burden estimation models: Quantify the number of individuals potentially infected with DR-mycobacteria. (7) Theoretical models: Theoretically explore interactions between susceptible and resistant strains. Note: "CE" stands for cost-effectiveness. For full details of aim and model type see S2 Text.

resistance was often found to be modelled alongside rifampicin and/or isoniazid resistance with only 3 models including resistance to all 4 first-line antibiotics, 2 with mono-resistances and 1 with a combination of all 4 resistances [33,37,42] (Fig 4).

41 theoretical models included resistance to a non-named antibiotic (S1 Table). One of these explored differences in drug action (bacteriostatic or bactericidal [48], and two explored antibiotic persistence [49,50] (S1 Fig). There were 38 theoretical modelling studies (S1 Fig) capturing "drug resistance", with four of these models exploring firstly hypothetical and then antibiotic-specific resistance (S1 Table).

## Stage 2 Results: Heterogeneous models

We found 23 models with bacterial heterogeneity—15 bacterial population and 8 between-host models (S2 Table) [8,20,33,34,37,43–45,48,49,51–63]. The distribution of model aims that these papers fall into were different from Stage 1 with 13 "parameter estimation", 8 "treatment interventions", 1 "theoretical", and 1 "non-treatment intervention". 12 of the 23 models modelled the immune system.

**Fig 4. Treemap of specific resistance classes included in models in stage 1 shows that the majority of models included MDR/RR and few included more than two resistance classes.** Each coloured cell represents a specific combination of resistances included in a model, with the size of the cell representing how many models included this combination of resistances. "Single" and "Multiple" sections refer to the number of antibiotic resistances included in a model, with "Multiple" referring to models that captured resistance to more than one antibiotic. "*" indicates the model included all possible combinations of antibiotic resistance listed. A = INH, RIF, MDR/RR, MOX, PZA, BDQ, PA, RIF + MOX, RIF + PZA, B = INH, RIF, MDR/RR, AMI, MOX, BDQ, RIF + MOX, RIF + AMI, RIF + BDQ, C = INH, RIF, MDR/RR, XDR, MDR + FQ, MDR + SLInject, D = INH, RIF, MDR/RR, XDR, Pre-XDR. Antibiotic abbreviations as follows: AMI = amikacin, BDQ = bedaquiline, CLR = clarithromycin, ETM = ethambutol, FQ = undefined fluoroquinolone, MOX = moxifloxacin, PA = pretomanid, PZA = pyrazinamide, STR = streptomycin, INH = isoniazid, RIF = rifampicin, MDR/RR = multidrug resistant/rifampicin resistant, XDR = extensively drug-resistant, SLInject = second line injectable antibiotic (from WHO guidelines 2014). S1 Fig shows all resistance categories per 195 models.

## Bacterial population models

The fifteen bacterial population models mostly captured multiple resistance classes (n = 13) (Fig 5 and S2 Table). One other considered a single resistance class of isoniazid only in an *M. tuberculosis* population and explored deterministically the impact of antibiotic exposure on resistance dominance with or without heterogeneity in fitness and mutation distributions [52]. Including heterogeneity in fitness and mutation distributions was also the most common method for exploring variation in models with multiple resistance classes. This was true both for stochastic and deterministic model structures [33,43,51,57,59,62], though one deterministic model only explored differences in mutation rates [43]. Four models additionally explored the impact of variation in growth rates induced by different metabolic states [20,34,45,60], with one model including fitness variation too [45].

Different clearance rates were used in 2 models, a PK/PD model and a bacterial dynamics model to differentiate between two resistant bacterial strains with the aim of determining the most effective treatment combination [48,58].

One model did not include AMR as a direct resistance to an antibiotic, but instead as persistence [49]. This was modelled as non-replicating bacterial populations and antibiotics had little to no effect on these bacterial populations. The model implemented heterogeneity by including fast and slow-growing bacteria.

## Between-host models

All eight between-host models were compartmental models. Six of these models explored the impact of including a distribution of fitness costs affecting transmission resulting from resistance-conferring mutations to prevalence of either a single [8,53,55,56] or multiple resistance classes [54,63]. Four of these six models were deterministic [53,55,56,63], with Knight et al. (2015) exploring a stochastic version in the supplementary materials [8]. Blower et al. (2004) explored a stochastic model that included heterogeneity by modelling strains of *M. tuberculosis* with different fitness rates but also cure, treatment, detection, and resistance mutation rates. The model aimed to estimate MDR-TB prevalence [54].

Two stochastic models were classified as heterogeneous as they included resistance compartments stratified with different resistant genotypes [44,61]. These papers had different aims: Kendall et al. [44] explored the impact of high and low levels of moxifloxacin resistance on treatment regimens and drug susceptibility testing. Pecerska et al. [61] estimated the fitness cost of MDR-TB with and without pyrazinamide resistance from a genetic data set.

## Use of data derived from the literature

All Stage 2 papers used at least one parameter sourced from existing literature, so no models were entirely theoretical. Some models used a primary data set that was collected from

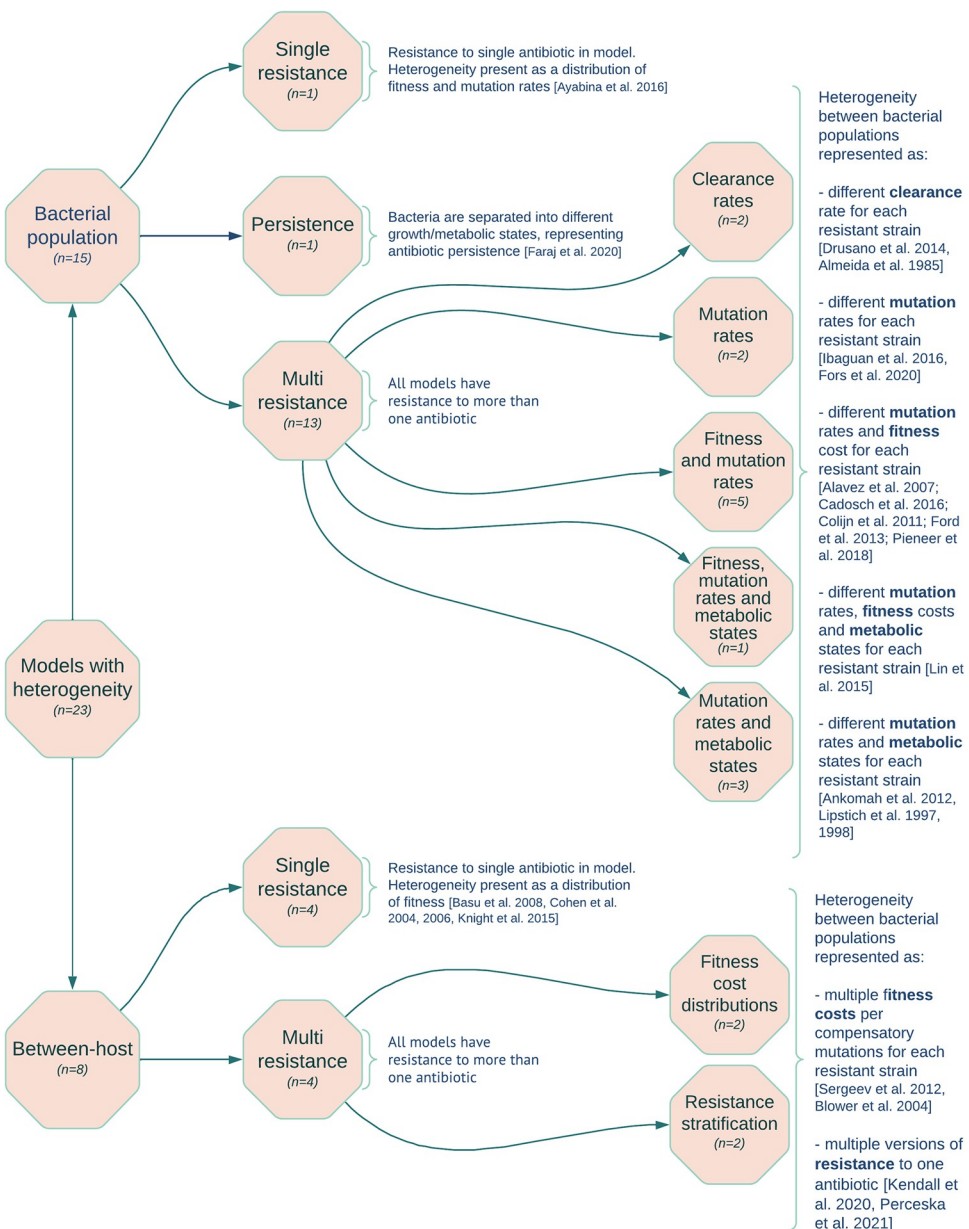

**Fig 5. Classifications of models including heterogeneity in their bacterial population (Stage 2) were split into bacterial population and between-host models and then stratified by whether they considered single or multiple resistance classes.**

experiments or a population study [20,49,52,58,59,61]. Data types used were experimental (83%), epidemiological (26%), clinical (4%), genetic (4%) and WHO data (30%). All bacterial population models used experimental data, with one paper also including clinical data [37]. Between-host models used a combination of experimental, epidemiological, and WHO data, with one using only genetic data.

## Acquired or primary resistance and discrete resistance

All models with heterogeneity represented resistance as discrete categories, such as MDR/RR-TB, with no models including resistance as a spectrum. 6/8 between-host heterogenous

models modelled resistance as both primary and acquired and two models had no primary resistance, with acquired resistance only [44,63].

## Resistance effects in models

Resistance affected the ability of *M. tuberculosis* to transmit in 6/8 between-host heterogenous models, with resistant strains usually having a lower value for the transmission coefficient or fitness parameter than the susceptible strain.

Resistance affected disease progression in all models except Knight et al. (2015) [8]. For bacterial population models, this was defined as different growth rates. For between-host models, this was included as a separate disease progression parameter for resistant strains [54,55,63], different relapse rates for patients with resistant bacteria [44], different associated mortality rates for each resistant strain [61], variance in cross-immunity by resistant strain [53], or different natural history pathways for resistant strains [56].

13/23 models assumed resistance affected operational parameters. In nine, resistance reduced treatment efficacy [8,44,45,53–56,61,63], with one also including different diagnostic (GeneXpert rapid nucleic acid amplification test for *M. tuberculosis*) sensitivity parameters for each resistant strain [44]. Four bacterial population models had a different antibiotic kill rate [48,49,58,60], with one including different clinical conversion factors [49].

## Discussion

Our review of the mathematical modelling landscape of drug resistance in mycobacteria has revealed a growing body of work mostly using transmission dynamic models to explore intervention impact. We found that a minority (33%) explore resistances other than MDR/RR-TB. Few models account for the known heterogeneity that exists in bacterial populations. Where heterogeneity was captured in both bacterial population and between-host models, it was mostly through a variation in the model-specific fitness parameter (with the definition of fitness varying broadly from being related to transmission, ability to cause disease or speed of bacterial growth).

Our Stage 1 landscape analysis found that several high MDR-TB burden countries (e.g. Pakistan, Nigeria, Ukraine, and Myanmar) are underrepresented in the English DR-TB literature. Increasing modelling of DR-TB in specific countries may aid understanding of epidemiology in the specific country and increase the global understanding of DR-TB, as well as improve estimates of intervention efficacy and hence design of context-specific interventions. This is highly relevant when considering that, as has been found for models of *M. tuberculosis* in general [11,64], most models aimed to estimate the impact of public health interventions. Transmission models were used more than any other type of model across all categories, except for the category of "treatment interventions + cost-effectiveness", where state transition models were most used. This indicates that most modellers are interested in modelling *M. tuberculosis* at a between-human host population scale.

MDR-TB was the most common category of resistance modelled (67% of DR-mycobacterium models)—an expected result linked to the historical importance of this as a clinical treatment threshold and reflected in most data collection [1,3]. Mono-isoniazid resistance was more commonly modelled than explicit mono-rifampicin resistance, with 27 models capturing the pathway from isoniazid resistance developing into MDR-TB. XDR-TB was not considered without MDR-TB other than by two papers by Basu et al. (2008, 2009), who were interested in the burden and interventions specific to XDR-TB [46,47]. XDR-TB was often treated as a final state of resistance in modelling systems, with no further resistance being acquired. This reflects the historic clinical decision-making pathway (susceptible or MDR or XDR) and that XDR-TB

is resistant to a large number of anti-TB antibiotics. However, there is a great variation in DR-TB and the pathways that may lead to each level of it. Understanding this variation in DR-TB will drive improvements in treatment success by identifying which antibiotics will be most effective and, therefore improve patient outcomes.

Rifampicin and isoniazid resistance were the most modelled mono-resistances, followed by pyrazinamide and ethambutol, reflecting first-line treatments and prophylaxis for TB and data availability. Testing for pyrazinamide and ethambutol resistance is typically reserved for reference settings, and there is widespread use of GeneXpert (Cepheid 6/10-colour instrument), which tests for rifampicin resistance. Only 21% of models (n = 41) captured resistances beyond these four drugs. This will need to be expanded as we move into a period with many more treatment options–constructing, parameterising, and exploring mathematical models of other antibiotic resistances is vitally needed to optimise future treatment and TB control interventions, as well as to explore evolutionary pathways. For example, we found only two papers which explicitly modelled resistance to bedaquiline [44,45], whilst two new treatment regimens containing bedaquiline were approved by the WHO in 2022 [65].

Models that capture non-specific DR-TB can be useful in the absence of data or to explore broad trends. We found 45 models in this category and found that these theoretical or non-specific systems were used to understand under what constraints DR-TB would dominate over DS-TB or explored the efficacy of a theoretical intervention.

When designing a model to answer a specific question such as the impact of a public health intervention, a balance needs to be struck between designing a detailed or generalised model to allow for a pragmatic approach. This pragmatism is likely the reason for our stage 2 results that revealed few models including bacterial heterogeneity. This is despite several models showing how heterogeneity in transmission fitness can affect DR-TB prevalence estimates [8,54–56]. Or how including multiple levels of resistance to one antibiotic can affect treatment outcomes [44,61]. Authors cannot capture all the subtlety of antibiotics as a selection pressure without including the related resistance dynamics and from this the population diversity it fosters. Mathematically, it can be difficult to include complexities in all aspects, for example, population mixing, and often there is little context-specific data on bacterial heterogeneity to inform models. However, if authors want to understand the risk of antibiotic resistance developing under a new treatment regimen it should follow that those resistances are then included in predictions. Some nuance may be beneficial in results that are only achievable with models that include bacterial heterogeneity, such as in Basu et al. (2008) where their conclusions suggested that a weaker immune response to a DR-TB infection with high fitness levels leads to higher DR-TB prevalence in HIV-positive and -negative populations [53].

Interestingly, we found that all models included resistance in a small number of discrete compartments, with no near-continuous distributions of resistance. Biologically speaking, resistance exists across a spectrum with strains having a range of minimum inhibitory concentrations, but for therapeutic and diagnostic uses they are classified with discrete values. Modelling resistance at multiple possible sub-levels would enable new research questions to be posed about pathways to evolution and competition due to multiple resistant levels. To our knowledge, such a question has not yet been asked regarding *M. tuberculosis*.

We found that transmission fitness levels, by contrast to resistance levels, were commonly allowed to vary across a distribution within resistant populations, likely reflecting the available historical data pointing to fitness differences between TB strains [66]. This contrasts with the lack of data linking resistant strain variation with treatment outcomes such as failure or recovery. Including such fitness effects is a relatively easy single-parameter effect within standard transmission dynamic or bacterial dynamics models and is commonly included in models of drug resistance outside of *M. tuberculosis* [7].

In this review, we identified 190 published papers which included drug-resistant strains of *M. tuberculosis*, a further 5 with a drug-resistant non-tuberculosis mycobacteria species, and 1 including both *M. tuberculosis* and *M. marinum*. Our update on the literature shows an increasing trend to model DR-TB.

The limitations of our review included that we conducted the search for English language articles when a substantial burden of DR-TB is found in non-English speaking settings such as Eastern Europe [1]. We did not capture which antibiotics were explored in the models as our focus was on the resistance captured nor time horizons for each model. Our stage 1 analysis only extracted high-level information as our main interest was the bacterial heterogeneity in stage 2. Future work could use this baseline set of literature to explore how resistance is modelled in the natural history of tuberculosis.

We encourage future modellers to consider if the bacterial component of their research question would benefit from the inclusion of bacterial heterogeneity. By not including it, models miss key features of bacterial populations, such as competition or treatment efficacy differences between strains and may, for example, under or overestimate the degree by which an intervention might increase resistance or prevalence of DR-TB.

We were unable to provide a comprehensive review of how resistance was included in Stage 1 models due to the lack of model information provided in many papers such as parameter tables, model diagrams or equations. Future mathematical models should aim for clear model reporting as suggested by the WHO [67] and Bennett et al. (2012) for transparency and to enable reproducible research [68].

In this review, we identified 195 drug-resistant mycobacteria mathematical models, with 190 DR-TB models and 23 models including bacterial heterogeneity. This has provided us with an understanding of how resistant mycobacterial species have been modelled, in terms of geographical settings, model aims and types, resistances modelled and further insights into the inclusion of bacterial heterogeneity. However, we found that bacterial heterogeneity was often ignored despite evidence of its importance at the population level. Balancing pragmaticism with biological reality when building mathematical models is vital within the fundamental evolutionary dynamics of AMR.

## Supporting information

**S1 Text. Search strings.**
(DOCX)

**S2 Text. Details of extraction table for stage 1.**
(DOCX)

**S3 Text. Details of extraction table for stage 2.**
(DOCX)

**S1 Fig. Heatmap of all resistance categories in stage 1 models.** Heatmap of resistances included per DR-TB model (n = 195) indicates a lack of diversity in resistances modelled, with MDR/RR-TB featuring in over half of all 195 models. Each coloured line indicates a model (y axis) included in stage 1 (purple) or stage 2 (orange). The graph groups models into specific (captures resistance to a named antibiotic), non-specific (defined resistance that are not specific to an antibiotic) and hypothetical (captures antibiotic resistance not linked to a named drug). Antibiotic acronyms as follows: AMI = amikacin, BDQ = bedaquiline, CLR = clarithromycin, ETM = ethambutol, FQ = undefined fluroquinolone, LZD = linezolid, MOX = moxifloxacin, PA = pretomanid, PZA = pyrazinamide, STR = streptomycin, INH = isoniazid, RIF = rifampicin, MDR/RR = multidrug resistant/ rifampicin resistant,

XDR = extensively drug-resistant, SLInject = second line injectable antibiotic (from WHO guidelines 2014), another 1st line = rifampicin, ethambutol, or pyrazinamide. Index links to paper number in S1 Table.
(DOCX)

**S2 Fig. Plot of number of publications over time.**
(TIF)

**S1 Table. Extraction table results from stage 1.**
(XLSX)

**S2 Table. Extraction table results from stage 2.**
(XLSX)

**S3 Table. Geographic settings in models.**
(DOCX)

## Acknowledgments

We would like to thank the support of library staff at the London School of Hygiene and Tropical Medicine. Thank you for the guidance and advice for this work from Quentin Leclerc and Alastair Clements.

## Author Contributions

**Conceptualization:** Naomi M. Fuller, Timothy D. McHugh, Gwenan M. Knight.

**Data curation:** Naomi M. Fuller, Christopher F. McQuaid, Martin J. Harker, Chathika K. Weerasuriya, Gwenan M. Knight.

**Formal analysis:** Naomi M. Fuller, Gwenan M. Knight.

**Investigation:** Naomi M. Fuller, Gwenan M. Knight.

**Methodology:** Naomi M. Fuller, Gwenan M. Knight.

**Project administration:** Naomi M. Fuller.

**Supervision:** Gwenan M. Knight.

**Validation:** Naomi M. Fuller, Christopher F. McQuaid, Gwenan M. Knight.

**Visualization:** Naomi M. Fuller, Gwenan M. Knight.

**Writing – original draft:** Naomi M. Fuller, Gwenan M. Knight.

**Writing – review & editing:** Naomi M. Fuller, Christopher F. McQuaid, Martin J. Harker, Chathika K. Weerasuriya, Timothy D. McHugh, Gwenan M. Knight.

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
