## [Decision Letter · Decision Letter 0]

16 Oct 2023

Dear Dr Fuller,

Thank you very much for submitting your manuscript "Mathematical models of drug resistant tuberculosis show little consideration of bacterial heterogeneity: a systematic review" for consideration at PLOS Pathogens. As with all papers reviewed by the journal, your manuscript was reviewed by members of the editorial board and by several independent reviewers. It's evident that the the systematic review is timely, yet it's evident that a strong justification of 'how', 'why' and 'when' models should account for bacterial heterogeneity is required. In light of the reviews (below this email), we would like to invite the resubmission of a significantly-revised version that takes into account the reviewers' comments.

We cannot make any decision about publication until we have seen the revised manuscript and your response to the reviewers' comments. Your revised manuscript is also likely to be sent to reviewers for further evaluation.

Sincerely,

Mark R. Davies, Ph.D

Academic Editor

PLOS Pathogens

Debra Bessen

Section Editor

PLOS Pathogens

Kasturi Haldar

Editor-in-Chief

PLOS Pathogens

orcid.org/0000-0001-5065-158X

Michael Malim

Editor-in-Chief

PLOS Pathogens

orcid.org/0000-0002-7699-2064

Reviewer's Responses to Questions

**Part I - Summary**

Reviewer #1: This paper provides a systematic overview of TB models which address questions related to drug resistance. The authors include a 2-stage review where they first identify within-host models and transmission models that consider questions related to drug resistance, then identify the subset of these models that allow for variation of resistant isolates (for the within-host models) or patients with resistant TB (for the transmission models).

The manuscript is useful in that provides a timely summary of the literature on modeling of DR-TB, with a specific question about how bacterial heterogeneity is handled (or omitted) from these models. A major challenge involved in this review is for the authors to provide clear description of the rationale for inclusion/exclusion of studies, the classification of these studies, and what is actually meant by “fitness” as this is a composite characteristic that may be modeled with a variety of mechanisms. I have some specific comments related to opportunities to improve these aspects of the paper below.

Further, I think the authors may want to soften the title and some of the language a bit. I agree with the authors that there may be important consequences for omitting consideration of such bacterial variation, but I do not think that every model-based analysis of drug resistant TB necessarily needs to include this type of variation to be useful. I suspect the authors agree with this, but the title sounds unnecessarily judgemental.

Reviewer #2: This manuscript aims to summarize the approaches used by published mathematical models of human mycobacterial infections (with a focus on tuberculosis) to evaluate whether and how they account for bacterial heterogeneity. This is an impressively comprehensive review of modeling studies, spanning the range from models of bacterial populations to decision-analytic models to evaluate the cost-effectiveness of tuberculosis treatment interventions. That alone makes this work highly valuable for tuberculosis researchers, and modelers in particular. There are a few key aspects of the work that could be improved to make it suitable for publication.

**Part II – Major Issues: Key Experiments Required for Acceptance**

Reviewer #1: Major comments:

1) Lines 131-135 and Lines 171-175 - I am not quite clear on the distinction being made between the types of heterogeneity considered for the i) within-host models vs. ii) transmission models. I assume the goal is to include models of each type when there is structure that allows the same pressure (ie drug treatment) to exhibit variable selection on mycobacteria of the same resistance phenotype (for within-host models) or individuals infected by mycobacteria with the same resistance phenotype (for transmission models)? In my view, being upfront about the ingredients needed to model selection of more fit DR variants provides a clear rationale for why some studies are included and others are not.

2) Relatedly, some of the wording is ambiguous in these sections and should be clarified, for example:

a. Line 133: “strains resistance (sic) to a single drug should also vary in another characteristic.” I assume this should be replaced by individuals infected with strains resistant to the same drug should also vary in another characteristic – that is, you are not interested in models that include INH and Rif monoresist TB of different transmission fitness, but rather models that include Rif resistant TB of variable transmission fitness.

b. Line 134: “multiple resistant strains were categorized and parameterized differently from eachother.” I assume this means that bacterial isolates of the same/similar resistance phenotype differed in way (e.g. growth rate)

3) For stage 1 review, I am wondering if the authors collected information about the time horizon modeled in each of the included studies (especially for the transmission models). Given the relatively slows dynamics of TB and DR-TB, I think readers would be interested in understanding the scope of the time horizons that have been modeled.

4) Figure 3: The categories of model types and aims for Figure 3 are confusing. Some of the type categories refer to the scale of the model (e.g. bacterial population) while some refer to the modeling approach (e.g. decision analytic), and some are subcategories of each other. The categories of model aims also seem a bit ad hoc and unclear and potentially overlapping (e.g. burden, data analysis vs theoretical, intervention). I find this difficult to make sense of and wonder if there is a more coherent way to categorize model types and usages for a reader such that the categories are mutually exclusive groups?

5) Figure 4: I understand the full matrix showing which papers included which resistance phenotypes is provided as a supplement and is probably too large to include in the main text as it is currently shown. But featuring these small subset of outliers with a large number of DR phenotypes in the main text (Figure 4 )doesn’t seem that helpful if the goal is to give the reader a fair sense for the scope of the literature. Wouldn’t it be possible to make a heatmap figure (or similar) that shows the frequency at which different DR phenotypes are included in all models in stage 1 (not just those with >3 pheotypes) even if you can’t show all of the details about the studies in the figure?

6) Line 406-7: “Where heterogeneity was captured, it was mostly through fitness variation”. I am unsure as what is really meant here as fitness is a composite characteristic. This is too vague a term and may differ for within host (eg differences in growth rates in absence of antibiotic pressure?) and transmission models (eg differences in transmission capacity?, differences in ability to cause disease after infection?).

7) I have questions about the completeness of the literature search (or maybe don’t quite understand the inclusion criteria), for example work from Abel zur Wiesch seems like it should have been included?

a. https://pubmed.ncbi.nlm.nih.gov/25972005/

8) I think there is a missed opportunity for the authors to use the Discussion to be more explicit about the consequences of not considering bacterial heterogeneity in these types of models – the decision (implicit or explicit) to omit this type of heterogeneity means that the models don’t allow for drug treatment to operate as a selective pressure. Whether this is a problem for models depends on the specific questions being asked, the specific selection pressure being exerted, and the time frame over which the effects are considered. The title of the paper suggests that this is almost always a problem, but I think worth considering being a bit less prescriptive. As the authors rightly argue elsewhere, all these models need to make simplifying decisions, so I would tend to be careful to suggest that models of DR TB always need to include such heterogeneity.

Reviewer #2: 1. What seems to be missing in the paper is a strong justification of why it matters to account for bacterial heterogeneity in models of tuberculosis. The authors mention that predicted outcomes may differ between models that do or do not account for heterogeneity but this is somewhat of a truism (we would expect that models with different structure and/or parameters would have different results). It would be helpful to provide more specifics in terms of the hypothesized differences (e.g., would we expect such models that do not include bacterial heterogeneity to underestimate or overestimate the development of resistance or effectiveness of treatments? Does the expected magnitude of these differences justify increasing model complexity?). If there is little evidence to date as to the value of incorporating bacterial heterogeneity in these models, than that can be stated as well, and can help to justify the need for this review.

2. Methods, line 132-135: can you please explain why bacterial heterogeneity was defined differently for bacterial vs. human population models?

**Part III – Minor Issues: Editorial and Data Presentation Modifications**

Reviewer #1: Minor comments:

1) Line 92 – the MDR/RR designation also reflects that the scale up of Xpert, which reports Rif-R resistance but not INH-R, has meant that we often know Rif-R status but not INH-R.

2) Line 198 – for the second stage of literature review, how many reviewers also help to identify which of the papers met heterogeneity inclusion criteria?

3) Line 460 – “mathematically, it is difficult to include complexities in all aspects…” I don’t think it is the math that makes it challenging to include complexity.

Reviewer #2: 1. It would be helpful to state clearly which aspects of bacterial heterogeneity the authors analyze in this work (e.g., heterogeneity in fitness cost of specific mutations, persistence in anatomic compartments, response to specific drugs). Stating this clearly in the Introduction will help keep the reader focused on what the primary goals of the paper are.

2. Please consider moving some of the Results section to the appendix so that the Results focus on the main purpose of the paper, to evaluated the inclusion of bacterial heterogeneity. There is lots of other very valuable information (e.g., study settings, data sources, etc…) but presenting the results in a kind of laundry list raises the risk of readers getting lost. At the very least, I would recommend placing the results on bacterial heterogeneity first, followed by all of the other results.

3. In the Discussion the authors bring up the fact that most papers incorporate resistance as a discrete value, but use continuous distributions for fitness costs. One potential reason for that is that there are point estimates for response to treatment conditional on drug resistance available in the literature or from primary data, whereas such data are rare if at all available for fitness costs (thus forcing modelers to consider a broad range of potential values across a continuous distribution).

4. The wording used in the paper can sometimes be vague or confusing. For example, rather than referring to “different resistances”, I would suggest “types” or “patterns” of resistance. Similarly, in the Supplement, I would suggest rephrasing “actual” vs. “theoretical” resistance to “specific” vs. “hypothetical”.

5. Please consider rephrasing “data analysis” to “parameter estimation” when describing the aims of models.

6. Results, line 227: the authors state that the distribution of countries covered in the selected modeling papers does not reflect the settings with highest DR-TB incidence but this is not really true: The countries with the most modeling papers are all in the top 5 of DR-TB incidence.

7. In the legend to Figure 4, please specify that the papers listed are those considering resistance to 3 or more specific drugs (rather than just “multiple”). Also, please make sure to include the paper indices matching the Table S1 in the figure (as mentioned in the legend).

8. Please review the supplemental materials carefully for typographical and grammatical errors.

PLOS authors have the option to publish the peer review history of their article (what does this mean?). If published, this will include your full peer review and any attached files.

Reviewer #1: No

Reviewer #2: No
---

## [Decision Letter · Decision Letter 1]

12 Feb 2024

Dear Dr Fuller,

Thank you very much for submitting your manuscript "Mathematical models of drug-resistant tuberculosis lack bacterial heterogeneity: a systematic review" for consideration at PLOS Pathogens. As with all papers reviewed by the journal, your manuscript was reviewed by members of the editorial board and by several independent reviewers. The reviewers appreciated the attention to an important topic. Based on the reviews, we are likely to accept this manuscript for publication, providing that you modify the manuscript according to the review recommendations.

The reviewers acknowledge substantial improvements to the manuscript. However, some minor comments have been raised that should be addressed.

Sincerely,

Mark R. Davies, Ph.D

Academic Editor

PLOS Pathogens

Debra Bessen

Section Editor

PLOS Pathogens

Michael Malim

Editor-in-Chief

PLOS Pathogens

orcid.org/0000-0002-7699-2064

Reviewer Comments (if any, and for reference):

Reviewer's Responses to Questions

**Part I - Summary**

Reviewer #1: (No Response)

Reviewer #2: The authors have appropriately addressed most of the comments from the first round of reviews. Again, this is an impressive and valuable review of approaches used to model durg resistance in mathematical models of tuberculosis. I only have minor editorial comments to improve the clarity for readers.

**Part II – Major Issues: Key Experiments Required for Acceptance**

Reviewer #1: (No Response)

Reviewer #2: (No Response)

**Part III – Minor Issues: Editorial and Data Presentation Modifications**

Reviewer #1: It does not seem consistent to list MDR/RR under the "multiple" drug resistant category as depicted in Figure 4, but then list models that only modeled MDR/RR as single resistance in Figure 5.

Perhaps I have misunderstood? But if not, this classification should be made consistent throughout the manuscript.

Reviewer #2: 1. I think that the manuscript would benefit from more clarity in the language used, especially when referring to drug resistance. Often the authors use terms such as "mutiple resistances". I would suggest selecting an alternative such as "classes of resistance", "categories of resistance" or "resistance phenotypes", and using it consistently throughout.

2. Similarly, the authors could consider distinghuishing the two broad classes of models that they considered as "bacterial population" vs. "human population".

PLOS authors have the option to publish the peer review history of their article (what does this mean?). If published, this will include your full peer review and any attached files.

Reviewer #1: No

Reviewer #2: No

Figure Files:

Data Requirements:

Reproducibility:

References:

---

## [Editor Report · Decision Letter 2]

25 Mar 2024

Dear Dr Fuller,

We are pleased to inform you that your manuscript 'Mathematical models of drug-resistant tuberculosis lack bacterial heterogeneity: a systematic review' has been provisionally accepted for publication in PLOS Pathogens.

Best regards,

Mark R. Davies, Ph.D

Academic Editor

PLOS Pathogens

Debra Bessen

Section Editor

PLOS Pathogens

Michael Malim

Editor-in-Chief

PLOS Pathogens

orcid.org/0000-0002-7699-2064

Minor comments from reviewers have been adequately addressed.
---

## [Editor Report · Acceptance letter]

3 Apr 2024

Dear Ms Fuller,

We are delighted to inform you that your manuscript, "Mathematical models of drug-resistant tuberculosis lack bacterial heterogeneity: a systematic review," has been formally accepted for publication in PLOS Pathogens.

Best regards,

Michael Malim

Editor-in-Chief

PLOS Pathogens

orcid.org/0000-0002-7699-2064